**Peer Review** The peer review history for this article is available as a PDF in the Supporting Information.

**Key Points:**

- Multiple wisps induced by NWC transmitter signals were captured by CIRBE/REPTile-2
- Simulations reproduce multiple wisps by including 1, −1, and 2 cyclotron resonance orders
- Highly oblique transmitter signals are necessary to produce multiple wisps

**Supporting Information:**

Supporting Information may be found in the online version of this article.

**Correspondence to:**

X. Li,
xinlin.li@lasp.colorado.edu

**Citation:**

Xiang, Z., Li, X., Baker, D. N., Mei, Y., O'Brien, D., Hogan, B., et al. (2024). Earth-based transmitters trigger precipitation of inner radiation belt electrons: Unveiling observations and modeling results. *AGU Advances*, 5, e2024AV001354. https://doi.org/10.1029/2024AV001354

**Author Contributions:**

**Conceptualization:** Xinlin Li
**Data curation:** Xinlin Li, Declan O'Brien
**Funding acquisition:** Xinlin Li
**Investigation:** Zheng Xiang, Yang Mei, Benjamin Hogan, Hong Zhao, David Brennan, Michael A. Temerin
**Methodology:** Zheng Xiang, Xinlin Li, Yang Mei, Hong Zhao, Binbin Ni, Theodore Sarris, Michael A. Temerin
**Project administration:** Xinlin Li, Daniel N. Baker
**Software:** Zheng Xiang, Declan O'Brien
**Supervision:** Xinlin Li, Daniel N. Baker
**Validation:** Zheng Xiang, Xinlin Li, Hong Zhao, Michael A. Temerin

# Earth-Based Transmitters Trigger Precipitation of Inner Radiation Belt Electrons: Unveiling Observations and Modeling Results

Zheng Xiang[1] , Xinlin Li[1,2] , Daniel N. Baker[1] , Yang Mei[1,2] , Declan O'Brien[1,2] , Benjamin Hogan[1,2] , Hong Zhao[3] , David Brennan[1] , Binbin Ni[4] , Theodore Sarris[1,5] , and Michael A. Temerin[6]

[1]Laboratory for Atmospheric and Space Physics, University of Colorado Boulder, Boulder, CO, USA, [2]Department of Aerospace Engineering Sciences, University of Colorado Boulder, Boulder, CO, USA, [3]Department of Physics, Auburn University, Auburn, AL, USA, [4]Department of Space Physics, School of Electronic Information, Wuhan University, Wuhan, China, [5]Department of Electrical Engineering, Democritus University of Thrace, Xanthi, Greece, [6]Retired from Space Sciences Laboratory, University of California, Berkeley, Berkeley, CA, USA

**Abstract** Human activity influence Earth's environment, including the space environment hundreds to thousands of kilometers above the Earth. One direct evidence is that the 19.8 kHz electromagnetic signals launched by the North West Cape (NWC) transmitter station in Australia produce a wisp-like energy distribution of precipitating energetic electrons in Earth's inner radiation belt, observed by many Low Earth Orbiting satellites. Typically, satellites observe a single wisp with energy that decreases with increasing L (approximately the radial distance in the equatorial plane), which is produced by a first-order cyclotron resonance of transmitter signals with energetic electrons. Here we report, for the first time, multiple wisps observed by the Relativistic Electron and Proton Telescope integrated little experiment-2 (REPTile-2) on board the recently launched Colorado Inner Radiation Belt Experiment (CIRBE) CubeSat. Based on simulation results, we demonstrate that harmonic cyclotron resonances including the 1, −1, and 2 orders by highly oblique NWC transmitter signals produce these multiple wisps. The discovery of multiple-order cyclotron resonances simultaneously occurring in space sheds new light on wave-particle interactions in near-Earth space. It also has implications for developing artificial radiation belt remediation techniques and understanding the propagation and scattering of plasma waves in planetary magnetospheres.

**Plain Language Summary** To communicate with submarines, very low frequency (VLF, 3–30 kHz) transmitter stations were built around the world since their signals can penetrate seawater to a few tens of meters. Generally, these VLF transmitter signals propagate within the Earth-ionosphere waveguide. On the nightside, electron densities in the ionosphere are weak, some transmitter signals can leak into the magnetosphere and interact with energetic electrons through cyclotron resonances. Previous satellite observations suggest that the cyclotron resonance between transmitter signals and energetic electrons is mostly first-order and produces a wisp-like energy spectrum of precipitation electrons. Thanks to the high-energy and time resolution of the REPTile-2 instrument, multiple wisps were captured and reproduced by simulations with harmonic resonance orders (−1, 1, and 2). The scattering effect induced by space plasma waves from different resonance orders has never been examined by observation until our present work. Energetic electrons in Earth's radiation belts have detrimental effects on spacecraft subsystems and are harmful to astronauts during extravehicular activity. Scattering by the high-power VLF waves has been considered a promising method for human control of Earth's radiation belts. Our results demonstrate that oblique human-made VLF waves can scatter electrons in a wider energy range through harmonic cyclotron resonances. This sheds new light on developing artificial radiation belt remediation techniques to mitigate radiation hazards on spacecraft and astronauts.

## 1. Introduction

Energetic electron fluxes in Earth's inner radiation belt increase during active geomagnetic times followed by slow decay during quiet periods (Selesnick, 2015a, 2015b; Xiang, Li, Temerin, et al., 2020). Their energy spectra can often be characterized by exponential distributions (Mihalov & White, 1966; Zhao, Johnston, et al., 2019). Due to atmospheric collisions and wave-particle interactions, trapped inner belt electrons can be scattered into the drift loss cone and precipitate after encountering the South Atlantic Anomaly (SAA), where the weaker magnetic

**Visualization:** Zheng Xiang, Yang Mei, Declan O'Brien, David Brennan, Theodore Sarris
**Writing – original draft:** Zheng Xiang
**Writing – review & editing:** Xinlin Li, Daniel N. Baker, Yang Mei, Declan O'Brien, Benjamin Hogan, Hong Zhao, David Brennan, Binbin Ni, Theodore Sarris, Michael A. Temerin

field leads to an altitude decrease of the electron's mirror point (Selesnick et al., 2003; Tu et al., 2010). The electrons in the drift loss cone are called quasi-trapped since their lifetime is shorter than one drift period, which depends on the energy and L of the electrons (e.g., a 500 keV electron at $L = 1.2$ has a drift period of 1.5 hr). Different from trapped electrons, energy spectra of quasi-trapped inner belt electrons typically follow a CRAND (Cosmic Ray Albedo Neutron Decay) -produced spectrum discovered by measurements from CSSWE/REPTile and DEMETER/IDP (Li et al., 2017; Selesnick, 2015a, 2015b; Xiang et al., 2019; Zhang et al., 2019). However, peaks in the energy spectra of quasi-trapped electrons are frequently observed at $L \approx$ 1.4–1.8 (Datlowe, 2006; Liu et al., 2022; Vampola & Kuck, 1978). The energy of the peaks decreases rapidly with increasing L. They were first observed by satellite 1971 089A (Imhof et al., 1973) and were first called a "wisp" to describe enhancements of quasi-trapped electrons induced by NWC transmitter observed by DEMETER/IDP (Sauvaud et al., 2008).

Energetic electrons trapped by Earth's magnetic field experience three periodic motions: gyration around a magnetic field line, bounce along a magnetic field line between mirror points, and azimuthal drift around the Earth. Associated with each periodic motion, plasma waves with similar frequencies (including Doppler-shifted effects) can resonate with energetic electrons, leading to changes of their energy, pitch angle, and L, accounting for the dynamics of Earth's radiation belts (Baker et al., 2013, 2014; Li & Hudson, 2019; Mann et al., 2016; Ripoll et al., 2020; Shprits et al., 2013, 2016; Thorne et al., 2013; Zhao, Zhou, et al., 2019). One common and important wave-particle interaction in Earth's radiation belts is cyclotron resonance induced by whistler mode waves, producing both acceleration and precipitation of energetic electrons (Horne et al., 2005; Kasahara et al., 2018; Roth et al., 1999; Thorne et al., 2010). Whistler mode waves such as chorus waves and plasmaphereic hiss can be naturally generated (Bortnik et al., 2008; Meredith et al., 2006; Omura et al., 2008; Teng et al., 2023). Human-made Very-Low-Frequency (VLF) electromagnetic signals leaking into near-Earth space also propagate in the whistler mode and can interact with energetic electrons (Koons et al., 1981; Sauvaud et al., 2008). One direct observational evidence is the wisp spectrum observed by Low Earth Orbiting (LEO) satellites in Earth's inner belt (Liu et al., 2022; Shen et al., 2022; S. Zhao, Zhou, et al., 2019). Previous theoretical and observational studies demonstrated that first order cyclotron resonance of NWC transmitter signals with energetic electrons can account for the formation of wisps (Gamble et al., 2008; Selesnick et al., 2013; Usanova et al., 2022).

The cyclotron resonance condition between NWC transmitter signals and electrons can be expressed as (Koons et al., 1981):

$$\omega - k_\parallel v_\parallel = N\frac{\Omega}{\gamma}, \tag{1}$$

where $\omega$ is wave angular frequency, $k_\parallel$ is the parallel component of the wave number, $v_\parallel$ is the parallel component of the electron velocity, $\gamma$ is the Lorentz factor, $\Omega$ is the electron gyrofrequency, and $N = 0, \pm 1, \pm 2 \ldots$ represent the order of the resonance with $N = 0$ corresponding to the Landau resonance. Note that $\Omega$ does not include the sign of the charge, as a convention in this study, so that $N = 1$ is the principal resonance for whistler mode waves interacting with electrons, contrary to other studies for which $N = -1$ is the principal resonance (e.g., Albert, 2005; Glauert & Horne, 2005). We can see from Equation 1 that $v_\parallel$ and thus energy must decrease with increasing L, that is, with decreasing $\Omega$ since $k_\parallel$ changes slowly with increasing L. For parallel propagating waves only the $N = 1$ order is effective. But for energetic electrons and obliquely propagating waves where the electron gyro-radius is comparable to the perpendicular wavelength, the other orders can be effective in pitch-angle scattering energetic electrons. For whistler mode waves, a positive resonance order means the direction of motion of waves and electrons are opposite, while a negative resonance order means the same direction. Different resonance orders produce distinct scattering effects on electrons, which have been theoretically studied for many years (Abel & Thorne, 1998, 1999; Albert, 2012; Albert et al., 2016, 2020; Lyons et al., 1971; Lyons, 1974; Ni et al., 2011; Ripoll et al., 2014; Ross et al., 2019). However, the exact scattering effects from different resonance orders have never been examined by observations. For naturally generated waves, the scattering effects from different orders are mixed up due to their wide frequency range. In contrast, the nearly monochromatic NWC transmitter signal, bandwidth $19.8 \pm 0.1$ kHz, allows scattering effects from different orders to be clearly distinguished. In this work, we report, for the first time, multiple wisps observed by CIRBE/REPTile-2 and experimentally verify scattering induced by different resonance orders.

Figure 1 illustrates the formation of wisps observed by CIRBE/REPTile-2. Harmonic resonances between NWC transmitter signals and energetic electrons produce single, double, and multiple wisps close to the location of the

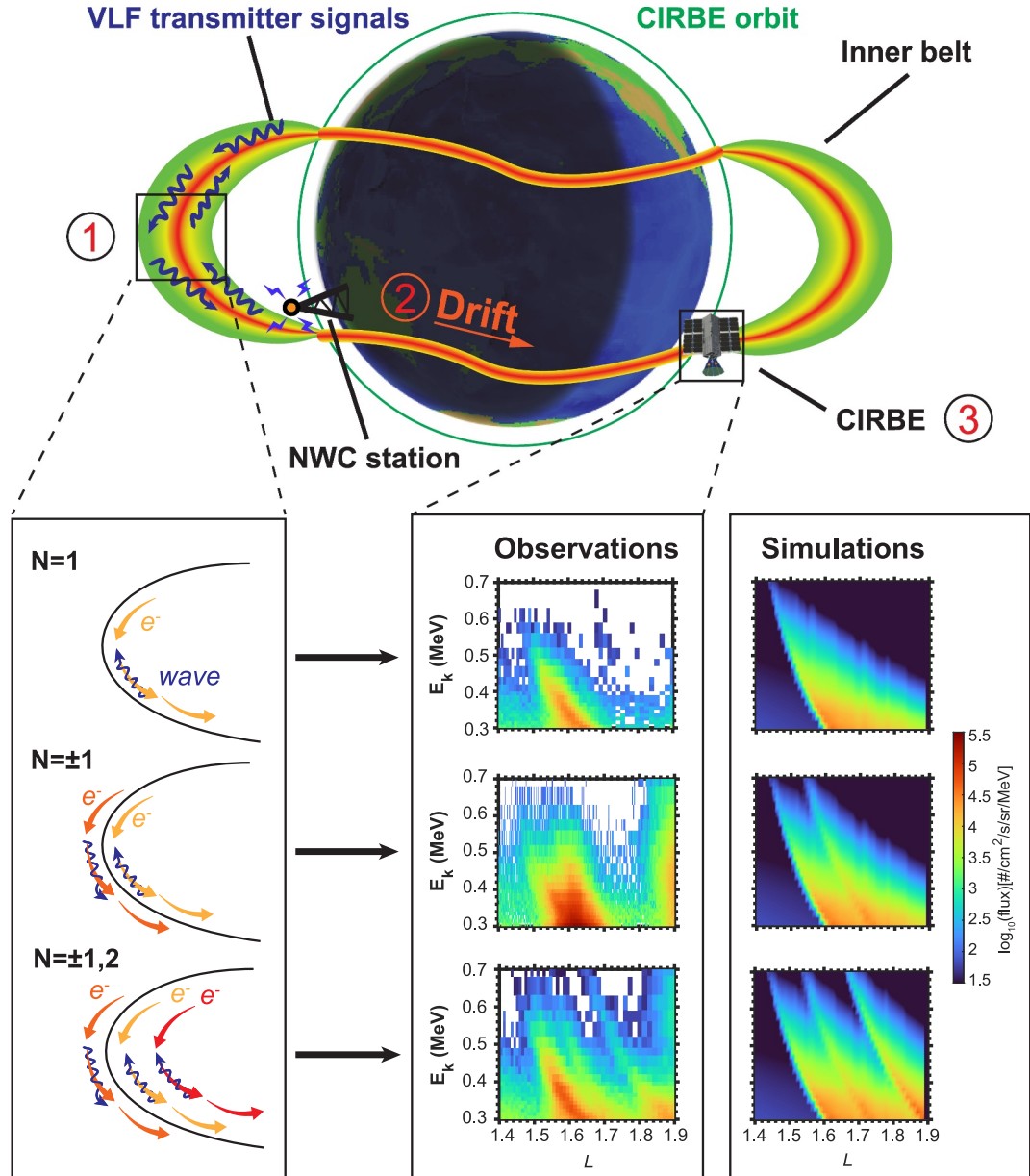

**Figure 1.** Schematic illustration of multiple wisps produced by harmonic cyclotron resonances of NWC transmitter signals with energetic electrons. On the nightside, signals launched by the ground based NWC station penetrate the ionosphere and leak into the magnetosphere. The thicker ionosphere on the dayside better shields the radiation belt from the NWC signal. These signals bounce between the northern and southern hemispheres to become highly oblique. The interaction between NWC transmitter signals and energetic electrons occurs at different energies through distinct resonance orders. For resonance order $N = 1$, the direction of motion of waves and electrons is opposite, while their direction is the same for $N = -1$. Similarly, waves and electrons move in opposite directions for $N = 2$ and the resonance energy is higher. Each resonance order produces a wisp signature in the electron energy spectra. Together, they produce multiple wisps (Step 1). In the zoom-in panel of Step 1, blue arrows indicate waves while yellow, orange, and red arrows indicate electrons with increasing energies. Wave-particle interactions only happen close to the NWC station due to the limited range of leaked signals in space. Then, the scattered electrons drift eastward (Step 2) until they precipitate into the SAA. CIRBE/REPTile-2 on the dayside observes multiple wisps drifting from the night-side (Step 3). The single, double, and triple wisps in the observation panels are from CIRBE/REPTile-2 data at 16:54–16:59 UT on 31 August, 16:15–16:20 UT on 15 December, and 16:29–16:33 UT on 28 August of 2023, respectively. During these periods, the NWC station was on the nightside. Simulations are from the Drift-Diffusion-Source model including the $N = 1$, $N = \pm 1$, and $N = \pm 1, 2$ resonance orders, respectively.

NWC station. These scattered electrons drift eastward due to the gradient and curvature of Earth's magnetic field and precipitate into the SAA where the weak magnetic field lowers the mirror point of the electrons. CIRBE/REPTile-2 observed these scattered electrons on the dayside with single, double, and multiple wisps due to scattering effects by $N = 1$, $N = \pm 1$, and $N = \pm 1, 2$ cyclotron resonances, respectively. The wisps from $N = 1$ and $N = -1$ partly overlap. Thus, fine energy and time resolution is needed to distinguish wisps from $N = 1$ and $N = -1$. This is likely why multiple wisps were overlooked by previous satellite measurements but captured by CIRBE/REPTile-2. When the NWC station is on the nightside and turns on, CIRBE/REPTile-2 almost always captures wisps to the west of the SAA. From the beginning of the mission to the end of year 2023 (256 days) within longitude range ($-140°$, $-70°$) in the southern hemisphere, CIRBE/REPTile-2 observed 105 single wisp events (59.7%), 22 double wisp events (12.5%), and 49 multiple wisp events (27.8%). The observed single- and multiple-wisp features are successfully reproduced in the simulation including different resonance orders with NWC transmitter signal, unveiling that the driving mechanism of multiple wisps are multiple harmonic cyclotron resonances. Note that the observed single wisps are produced by transmitter signals with smaller wave normal angles. Their scattering effects are mainly from the first-order resonance.

## 2. Observations

We obtained measurements of multiple wisps through the Relativistic Electron and Proton Telescope integrated little experiment-2 (REPTile-2) on board the Colorado Inner Radiation Belt Experiment (CIRBE), a 3U CubeSat launched into a sun synchronous orbit (97.4° inclination, 509 km altitude, with a local time ascending node at 10:30 and a descending node at 22:30) on 15 April 2023 (Li, 2024; Li et al., 2022, 2024). REPTile-2 incorporates pulse-height-analysis to enable 60 electron energy channels (0.25–6 MeV) and 60 proton channels (6.5–100 MeV) at a cadence of one second in the SAA regions and five seconds outside (Khoo et al., 2022; Li et al., 2024). The nominal energy resolution (ΔE/E) is ∼10% for the electron energy channels. Guard rings are used to eliminate signals from side penetrating particles. The active attitude control system on CIRBE keeps the pointing direction of REPTile-2 nearly perpendicular to the background magnetic field with a ∼51° field of view.

Using electron measurements with fine energy and time resolution from CIRBE/REPTile-2, we found five multiple-wisps events during 28–29 August 2023, consisting of three wisps with gaps in between. Geomagnetic activity was quiet during the 2 days with Kp ≤ 2 and Dst ≥ −12 nT. Figure 2a displays the satellite tracks corresponding to the electron fluxes shown in Figures 2b–2f. Each satellite track lasted around 4 minutes during which the NWC station was on the nightside. Electrons were measured to the west of the SAA above the southern hemisphere. All wisps have lower energies at higher L values. The three wisps cover $L = 1.45–1.65$, $L = 1.6–1.75$, and $L = 1.7–1.9$, respectively. Their energies cover 300–600 keV. The first and third wisps have larger fluxes than the second wisp. Flux levels in Figures 2b and 2d are higher since they are measured closer to the SAA where the magnetic field is weaker, and the observed electrons have higher equatorial pitch angles (see Figure S7 in Supporting Information S1). Electron fluxes at higher equatorial pitch angles have larger values (Datlowe, 2006; Shen et al., 2022) since wisps are formed by electrons scattered from the trapped population. For the same reason, wisps seen in the northern hemisphere have lower fluxes because the magnetic field is stronger there.

## 3. Simulations

We performed numerical simulations to test our hypothesis that multiple wisps are caused by harmonic cyclotron resonance of energetic electrons with NWC transmitter signals. We use the Drift-Diffusion-Source model (Liu et al., 2022; Xiang et al., 2019, 2020a, 2020b) to reproduce observations in Figures 2b–2f. Different from the traditional Fokker-Planck simulations adopted for Earth's radiation belts, the Drift-Diffusion-Source model also includes a drift term and uses the IGRF magnetic field model to calculate the bounce loss cone at different longitudes (Selesnick et al., 2003; Tu et al., 2010). Thus, the Drift-Diffusion-Source model can simulate the dynamics of quasi-trapped electrons and have reproduced the longitudinal distributions of quasi-trapped electrons produced by CRAND (Xiang et al., 2019) and NWC transmitter signals (Liu et al., 2022). The equation of the Drift-Diffusion-Source model is given by:

$$\frac{\partial J}{\partial t} + \omega_d \frac{\partial J}{\partial \phi} = \frac{1}{G_0(\alpha_{eq}) \sin(2\alpha_{eq})} \frac{\partial}{\partial \alpha_{eq}} \left[ G_0(\alpha_{eq}) \sin(2\alpha_{eq}) \langle D_{\alpha_{eq}\alpha_{eq}} \rangle \frac{\partial J}{\partial \alpha_{eq}} \right] + S_e, \tag{2}$$

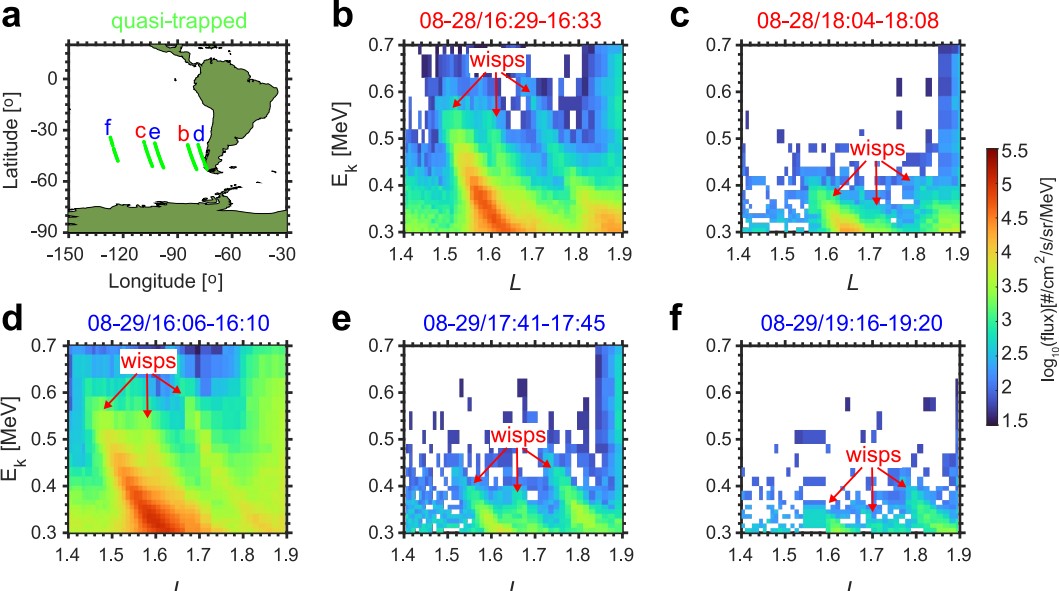

**Figure 2.** Multiple wisps observed by CIRBE/REPTile-2 on two successive days of 28–29 August 2023. (a) Satellite tracks of the five multiple wisps events in panels (b–f). The measured electrons are all quasi-trapped (with pitch angles larger than the local bounce loss cone but smaller than the drift loss cone, these electrons precipitate into the atmosphere when they drift into the SAA). Red and blue lettering indicate measurements on 28 and 29 August 2023, respectively. (b–f) Electron flux measurements as a function of energy and L, calculated from the IGRF magnetic field model. Three wisps shown in each panel are indicated by red arrows.

where $J$ is the electron flux, $\omega_d$ is the bounce-averaged drift frequency, $\phi$ is the drift phase or geomagnetic longitude, $G_0(\alpha_{eq}) = v\tau_b/2$ for the bounce period $\tau_b$ and the electron speed $v$, $\alpha_{eq}$ is the equatorial pitch angle, and $t$ is time. $S_e$ is the electron source rate from CRAND which is the main source of quasi-trapped electrons in the inner belt when the NWC station is on the dayside (Liu et al., 2022). $\langle D_{\alpha_{eq}\alpha_{eq}} \rangle$ are bounce-averaged pitch angle diffusion coefficients.

We use a standard fully implicit method to solve Equation 2. The grid points in $\alpha_{eq}$ and $\phi$ are $900 \times 360$. The empty bounce loss cone is indicated by $J = 0$ as a boundary condition. The other boundary condition is $\frac{\partial J}{\partial \alpha_{eq}} = 0$ at $\alpha_{eq} = 90°$ for each $\phi$. In the simulation, one time step equals the time for electrons to drift one degree in geomagnetic longitude. One simulation example for 300 keV electrons at $L = 1.6$ is shown in Figure S7 in Supporting Information S1.

Based on quasi linear theory, the scattering effects induced by NWC transmitter signal can be represented by pitch angle diffusion coefficients (Lyons, 1974; Lyons et al., 1971; Ni et al., 2011) used in the Drift-Diffusion-Source model. The initial electron flux (Figure 3a) is the trapped electron flux measured by CIRBE/REPTile-2 at 13:23–13:29 UT on 28 August 2023 (∼3 hr before the multiple wisps shown in Figure 2b). The variation of trapped electrons in the inner belt is slow during quiet periods. Thus, the same initial flux is used for the simulation in Figures 3b–3f. The simulation reproduces the three-wisp features with good agreement with L and energy, demonstrating that multiple wisps are caused by the harmonic cyclotron resonance of electrons with NWC signals. In addition to the wisp positions, flux levels of wisps are also well reproduced in the Drift-Diffusion-Source model. Figure 3d has the highest flux levels while Figure 3f has the lowest flux levels. This explains why CIRBE/REPTile-2 observe fewer wisps far away from SAA regions.

In the inner belt, since frequencies of NWC transmitter signals are well below the electron gyrofrequency (109.2 kHz at the equator and 1380.9 kHz at the surface of Earth for $L = 2$), and cyclotron interactions mainly produce pitch angle diffusion (Kennel & Engelmann, 1966). The Full Diffusion Code (Ni et al., 2008, 2011; Shprits & Ni, 2009) is used to compute the bounce-averaged pitch-angle diffusion coefficients based on the quasi-linear theory (Albert, 2007), in which the scattering effect of waves on electrons can be modeled as a diffusion equation (Kennel & Engelmann, 1966), with the diffusion coefficients proportional to the wave power. The Full

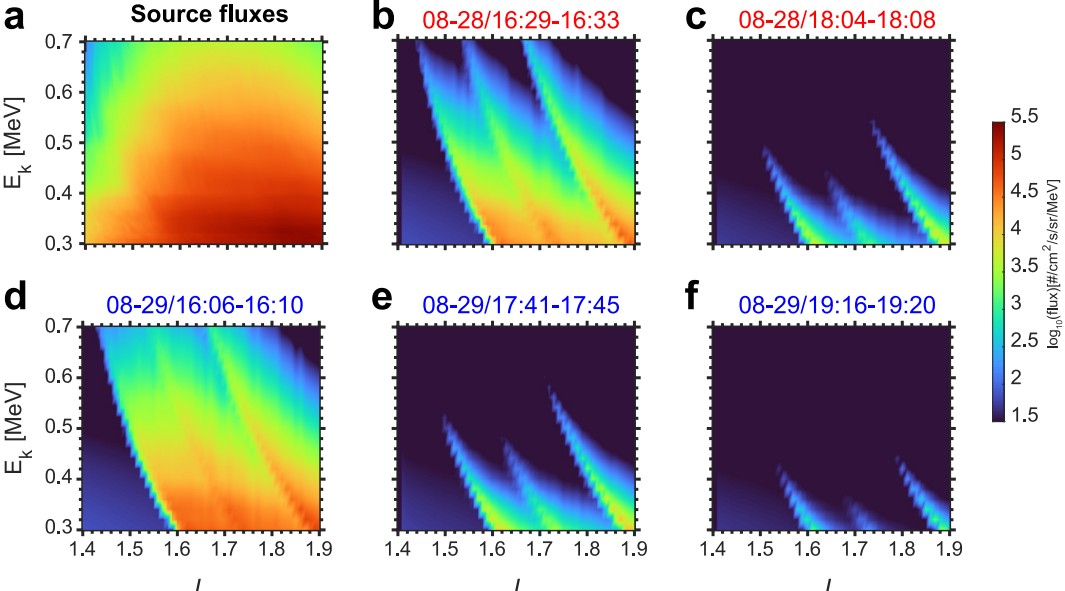

**Figure 3.** Simulation results of multiple wisps using the Drift-Diffusion-Source model. (a) Source flux used in the model to produce quasi-trapped electron fluxes scattered by NWC transmitter signals. (b–f) Simulated electron fluxes correspond to Figures 2b–2f. Scattering effects from $N = \pm 1, 2$ resonance orders are included in the simulations.

Diffusion Code calculate diffusion coefficients in energy and pitch angle given the magnetic field, plasma density, frequency, and the wave normal angle distribution of waves. It has been previously used to calculate diffusion coefficients induced by naturally generated and human-made whistler mode waves (Hua et al., 2020; Liu et al., 2022; Thorne et al., 2013; Xiang, Li, Ni, et al., 2020). To calculate the pitch-angle diffusion coefficients, NWC transmitter signals are assumed to have a Gaussian frequency distribution, centered at the frequency 19.8 kHz with a bandwidth of ±100 Hz. The wave amplitude is set as 20 pT. The wave normal angle distribution is set as θmin = 55°, θmax = 75°, θm = 65°, and θw = 10°. The centered wave normal angle in previous studies is usually set as θm = 45° (e.g., Abel & Thorne, 1998; Kim et al., 2011). Here, we use a more oblique wave normal angle distribution to obtain stronger scattering effects contributed from $n = -1$ and $n = 2$. Considering that the refraction index approaches infinity when the normal angles of highly oblique waves approach the resonance cone angle, we set the pitch angle diffusion coefficient to zero when the refraction index is larger than 200 (Albert, 2017). We assume that transmitter signals are present at $L = 1.4–1.9$ and are confined within the longitude range of (99°, 129°) and the latitude range of (−25°, 25°) following previous studies (Liu et al., 2022; Ma et al., 2017; Meredith et al., 2019). The background electron density is 1.5 times larger than the model proposed by Ozhogin et al. (2012). Higher electron densities lead to lower resonant energies. In contrast, higher magnetic fields produce higher resonant energies.

The pitch angle diffusion coefficients used in the simulation at six given L values are shown in Figure 4. The scattering effects from different resonance orders are evident in each panel. The resonance energy of electrons increases with resonance order $N = 1$, $N = -1$, and $N = 2$, respectively. At $L = 1.4$, the pitch angle diffusion coefficients from $N = -1$, and $N = 2$ are non-zero only for electrons with energies higher than 700 keV (Figure 4a). Thus, their effects are not shown. In addition, the diffusion coefficients from $N = 1$ are mainly in the drift loss cone (indicated by the white dashed lines), suggesting that few electrons can be scattered from the trapped population into the drift loss cone at $L = 1.4$. Accordingly, there is no wisp signature in simulation results at $L = 1.4$ (Figures 4b–4f). The resonant energy of electrons from different orders decreases at higher L, leading to lower energies of wisps at higher L (Figures 4b–4f). Diffusion coefficients from $N = 1$ and $N = 2$ are stronger than those from $N = -1$. As a result, the first and third wisps are stronger than the second wisp. We assume unducted VLF waves (center wave normal angle is 65°) in the simulation, which is consistent with previous ray-tracing calculations during quiet geomagnetic conditions (Abel & Thorne, 1998; Koons et al., 1981). Statistical studies also suggest that unducted signals have a dominant occurrence in the inner belt and ducted signals show a higher occurrence rate during/after magnetic active periods (Gu et al., 2021; Singh, 1976).

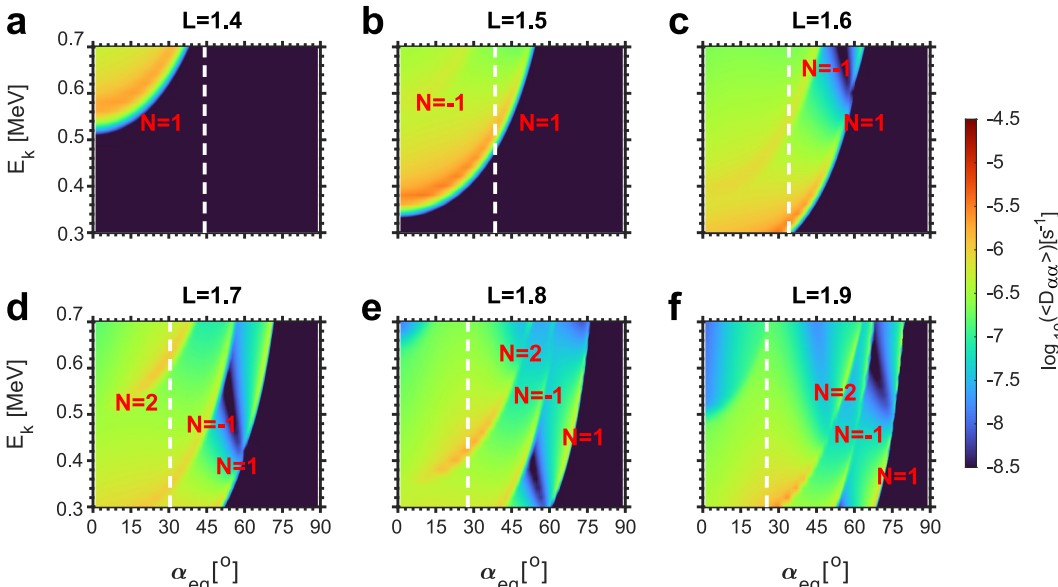

**Figure 4.** (a–f) 2-D plots of bounce-averaged pitch-angle diffusion coefficients $\langle D_{\alpha_{eq}\alpha_{eq}} \rangle$ as a function of electron kinetic energy and equatorial pitch angle at six selected L values. The drift loss cone at the given L is indicated by the white dashed lines. The diffusion coefficients from different resonance orders are indicated.

## 4. Discussion and Conclusions

Our analysis of multiple wisps measured by CIRBE/REPTile-2 demonstrates that these observations are produced by harmonic cyclotron resonances of VLF transmitter signals from the NWC station with energetic electrons. This is the first observational evidence of multiple harmonic resonances occurring simultaneously in space. Due to the narrow bandwidth of the NWC transmitter signals and the high energy resolution measurements of REPTile-2, scattering from different resonance orders is clearly distinguished in both observations and simulations. The pitch-angle diffusion coefficients (Figure 4) used in the simulation are calculated based on quasi-linear theory. The consistency between observations and simulations validates quasi-linear theory in quantifying the scattering effects of whistler mode waves. Although quasi-linear theory has been established for ~60 years in radiation belt physics (Kennel & Engelmann, 1966; Lerche, 1968; Lyons, 1974; Lyons et al., 1971; Summers et al., 2007) and has successfully explained many observational phenomena (Horne et al., 2008; Shprits et al., 2016; Thorne et al., 2013; Xiao et al., 2015; Zhao, Ni, et al., 2019), the exact values of diffusion coefficients are still under debate due to the complex calculation process and the numerous assumptions (Cunningham, 2023). Multiple wisps provide an excellent opportunity to examine the theoretical calculation of diffusion coefficients by different resonance orders.

Wisps can also be used to infer wave normal angles and amplitudes of NWC transmitter signals in the inner belt. Previous studies suggest that unducted VLF transmitter signals mainly occur at $L < 1.5$ and have weak scattering effects on energetic electrons (Clilverd et al., 2008; Gamble et al., 2008). Multiple wisps demonstrate that highly oblique transmitter signals certainly exist at wide L ranges, for example, $L = 1.4–1.9$, and produce significant scattering effects on energetic electrons. Parallel transmitter signals have no scattering effects at higher resonance orders ($N = -1, 2$). The comparison of pitch angle diffusion coefficients induced by transmitter signals with small and large wave normal angles are given in Figures S1–S3 in Supporting Information S1. Figures S4 and S5 in Supporting Information S1 show simulation results of wisps produced by transmitter signals with 40° and 15° center wave normal angle. Two wisps occur in Figure S4 in Supporting Information S1 and only one wisp is clearly observed in Figure S5 in Supporting Information S1, demonstrating the requirement of highly oblique transmitter signals in the formation of multiple wisps.

Using observations from the PROBA-V satellite, Cunningham et al. (2020) reported that the ratio of averaged flux computed with the NWC "on" versus "off" has a primary peak at lower L and a secondary peak at higher L. This phenomenon looks like a two-wisp event. Due to the rough energy resolution (seven channels for energy range

0.5–20 MeV), PROBA-V/EPT observations do not show clear wisp structures. Cunningham et al. (2020) calculate resonance energies of electrons for $N = 1$ and $N = -1$ at different wave normal angles to explain the observed primary ($N = 1$, parallel waves) and secondary ($N = -1$, oblique waves) peaks. In our study, we assume the same wave normal angles at different L values and only use different resonance orders ($N = -1, 1, 2$) to explain the observed three wisps. The assumed wave normal angles ($\sim 65°$) in our study are higher than those ($\sim 35°$) in Cunningham et al. (2020).

The amplitude of transmitter signals that leak into higher altitudes is also highly debated, ranging from a few pT to hundreds of pT (Koons et al., 1981; Ma et al., 2017; Shen et al., 2022). The multiple wisps on 28–29 August 2023 suggest that NWC transmitter signals in the inner belt are around 20 pT, which is comparable to previous inferred transmitter signal amplitudes ($\sim 15$ pT) from the longitudinal distribution of quasi-trapped electron fluxes scattered by NWC transmitter signals (Liu et al., 2022). Abel and Thorne (1998) combined eight most powerful VLF transmitters at the time into two frequency groups (17.1 and 22.3 kHz) and calculated their amplitudes based on the method of Inan et al. (1984). They suggested that the amplitudes of the 22.3 kHz signal group are 15–20 pT at $L = 1.5$–2, comparable to those inferred from multiple wisps. Note that the NWC transmitter was operating at 22.3 kHz at the time and had changed to 19.8 kHz when multiple wisps were observed by CIRBE/REPTile-2. The electron lifetime calculated in Abel and Thorne (1998) suggests that transmitter signals significantly influence energetic electron distributions at $L < 2.8$. However, Kim et al. (2011) found that transmitter signals do not contribute to the loss of relativistic electrons in the slot region using a much smaller wave amplitude (0.8 pT) in the 3D Versatile Electron Radiation Belt (VERB) code. Based on first-principles modeling of VLF transmitter from the source to the plasmasphere, Starks et al. (2020) obtained larger wave amplitudes of NWC transmitter signals (a few pT with longitude-averaged). It should be noted that the 20 pT amplitude of NWC transmitter signals inferred from multiple wisps are local wave intensities, namely not longitude averaged. To study long-term scattering effects induced by transmitter signals, the longitude dependence of signal intensities should be considered.

Similar to multiples wisps, "zebra stripes" in the inner belt also show multiple peaks in the spectrograms of energetic electrons (Lejosne & Roederer, 2016; Li et al., 2024; Ukhorskiy et al., 2014). However, there are several differences between the two phenomena: (a) zebra stripes are mainly observed in trapped electrons while multiple wisps are mostly observed in quasi-trapped electrons; (b) the flux peaks of zebra stripes at different L values follow the same drift period of electrons while wisps follow the cyclotron resonance energy curves induced by VLF transmitter signals; (c) the number of flux peaks in zebra stripes increase with time while multiple wisps do not have this temporal feature.

In theory, signals with different frequencies from three distinct VLF transmitter stations can also produce multiple wisps with only the $N = 1$ cyclotron resonance. When the wisps in Figure 2 were observed, together with the VLF transmitter station that is discussed above, NWC (19.8 kHz, 1000 kW, $L = 1.42$), there are two more VLF transmitter stations on the nightside (See Figure S3 in Supporting Information S1): JJI (22.2 kHz, 100 kW, $L = 1.25$) and unid25 (25 kHz, 250 kW, $L = 1.32$). However, we note that the transmitted power of the JJI and unid25 stations is much weaker (Meredith et al., 2019). In addition, their frequencies are higher than the NWC station, suggesting that the energy of electrons scattered by these two stations would be lower (Summers et al., 2007). Instead, we observe that the second and third wisps in Figure 2 have higher energies and weaker flux levels than the first wisp. Thus, we exclude the possibility that transmitter signals from different stations can produce the multiple wisps reported in this study.

Scattering by high-power VLF waves has been considered a promising method for controlling the radiation belts (Carlsten et al., 2019; Foster et al., 2016; Ni et al., 2022; Sauvaud et al., 2008). With harmonic cyclotron resonances, electrons scattered by human-made VLF waves expand to higher energies, leading to a stronger loss of energetic electrons in Earth's inner belt. Our results shed new light on developing artificial radiation belt remediation techniques to mitigate radiation hazards on spacecraft and astronauts.

## Conflict of Interest

The authors declare no conflicts of interest relevant to this study.

## Data Availability Statement

The CIRBE/REPTile-2 data are publicly available at https://lasp.colorado.edu/cirbe/data-products/.

**Acknowledgments**

We thank the entire CIRBE team. Especially, we owe our deepest gratitude to Rick Kohnert of LASP, who was the project manager for CIRBE and guided engineers and students since the "cradle" phase of CIRBE. Rick passed away before the launch of CIRBE, but we are certain that he would be proud of the contributions that CIRBE has made and will continue to make to the scientific community. This work was supported by the NASA Grants 80NSSC19K0995 and 80NSSC21K0583, and NSF Grant AGS 2348553.

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
