## [Peer Review History · Agu Advances]

Peer Review History for 2024AV001354

Reviewer #1

This is a very interesting paper presenting an observational evidence of the scattering by VLF transmitters and even demonstrating distinctive features of different harmonic scattering. The paper is well written and I only have minor comments.

Comments:

The oblique VLF transmitter scattering has been done in the past including resonances from -5 to +5 in Kim, et al. (2011), Understanding the dynamic evolution of the relativistic electron slot region including radial and pitch angle diffusion, *J. Geophys. Res.*, 116, A10214, doi:10.1029/2011JA016684.

I would also suggest to discuss that previous estimates may have underestimated the transmitter scattering as they used a correction according to Starks 2008

Starks, M. J., R. A. Quinn, G. P. Ginet, J. M. Albert, G. S. Sales, B. W. Reinisch, and P. Song (2008), Illumination of the plasmasphere by terrestrial very low frequency transmitters: Model validation, *J. Geophys. Res.*, 113, A09320, doi:10.1029/2008JA013112.

That estimate may have been underestimation as there is a broad illuminated area and the peak intensity may be missed by a spacecraft.

I would review more carefully the previous work and also discuss the estimates of Abel and Thorne and Startks et al

For the references to FDC I would suggest to cite the study of

Shprits, Y. Y., and B. Ni (2009), Dependence of the quasi-linear scattering rates on the wave normal distribution of chorus waves, *J. Geophys. Res.*, 114, A11205, doi:10.1029/2009JA014223.

Which was the first time the code was used for the radiation belts. That would also give a proper credit to both of the developers of the FDC code.

Reviewer #2

[Reviewer Comments begin on the next page.]

Review for JGR Advances
Rerence: 2024AV001354 (Xiang et al.)

The article “Earth-Based Transmitters Trigger Precipitation of Inner Radiation Belt Electrons: Unveiling Observations and Modeling Results” written by Xiang et al. and submitted to JGR Advances is about observations of electrons flux at LEO which present a typical “wisp” shape. Observations are made from CIRBE, a cube-sat built and managed by the authors’ team. This wisp has been observed in the past, first by Sauvaud et al. in 2008 from DEMETER. What is particularly interesting and important in CIRBE’s results is the observation of multiple wisps. These wisps have been attributed to the NWC VLF transmitter signals emitted from Australia in the past. Most of the article is focused on the authors’ attempt to explain the flux by modeling and simulation, with explanations that I question in following.

The observations of CIRBE of the wisp are very interesting, in particular their shape with multiple wisps and the variability among them. With proper discussion and comparisons to literature, this would be sufficient to make an interesting article, may be not in JGR Advances.

The main concerns I have are for the explanations given by the authors. The authors develop ideas based on which “harmonics” during wave-particle interactions would dominate and control the interaction. They argue that different resonance orders have never been examined by observations, claiming they can differentiate between harmonics. This can be questioned as the summation over all harmonics at a given location and energy should make this not possible, or hardly possible, likely not observable. I find their demonstration partially supported in this article. Questions are based on the fact that all harmonics contribute at a given energy and location and there is most of the time the domination of the $n=-1$ harmonic over all the others. Exceptions matter though, but would have to be proven. Generally, if conditions were such that a lower harmonic dominated the $n=-1$ harmonics, this would have to occur on a limited pitch angle range. There would be conditions, which would have to be discussed in the article. This would also be possible or favored with large wave normal angle. But high wave normal angle will in turn drastically reduce the scattering rate associated to wave particle interactions. As all harmonics shall contribute, Figure 1 sketch is doubtful: if the bottom panel is the results of all contributing harmonics ($|n|<3$), which we could believe, this would lead to three wisps occurring at 3 different L-shells (bottom panel) according to the authors, but how would the one single wisp configuration (top panel) possibly exist? Such configuration would be dominated by only one harmonic. But why would the other two wisps not exist since they were just proven to exist by the observation in the bottom panel?

Such discussion is absent of this article and lacking and conclusions are doubtful and drawn too quickly. Moreover, some part of this discussion has been published in Cunningham et al. GRL 2021, where these authors observe for the first time at LEO high flux of electrons above 500 keV, which they proved to be related to the NWC (by comparing on and off periods). These authors evaluate the resonance condition for the two first main harmonics (see their figure 2) and can this way relate flux, energy and location, while discussing the harmonics importance. In their discussion, they show how the harmonic number can influence the location, similarly to what is done here. There is some anteriority of this results. The Cunningham et al. 2021 GRL article is presently not cited by the authors, but cited in their previous articles (Liu et al., 2022; Li et al., 2024).

I list in following the major questions/comments which I recommend the authors perform to assess their claim before publication.

- The authors should demonstrate that, although harmonics accumulate at the same energy/location, they can be differentiated. That will require a discussion on the observed and computed pitch angle range (and wave normal). The author can choose a relevant energy at which they should plot the diffusion coefficients versus pitch angle for the following harmonics $n=-3 \dots n=0 \dots n=3$ as well as all harmonics summed, all plotted on the same plot. That will allow to show the dominance of one harmonic versus the others and the overall contribution of each within the sum, for a given pitch angle range. This should be done at relevant L-shell of the study, each indicative of a different wisp. I recommend a fixed energy of $E=(300, 500)$ keV and fixed L-shell $L=(1.5, 1.7, 1.9)$ with D_{aa} plotted versus pitch angle in 1D. This makes a figure with 6 panels, which can be put in the supplementary material.
- As explained above, increasing the wave normal angle will reduce the wave particle interactions. This effect has also a pitch angle dependence. A comparison between diffusion coefficient at large/small wave normal angle versus pitch angle at the above energy/location will allow a quantification of this effect. This makes another figure with 6 panels. The discussion in the text should indicate by how much the diffusion rate will be reduced by increasing the wave normal angle.
- The initial condition of the simulation is not plotted but should be.
- How long are run the simulations compared with a drift period? In particular, plotting results in terms of iteration steps is meaningless and should be converted in measure of the drift period.
- Which pitch angle range is observed, simulated, and plotted?
- The authors wrote “The Full Diffusion Code (Ni et al., 2008, 2011) is implemented”. A code is not “implemented”. A method is. Which code is actually used in this study?
- The authors should give the sign convention they use for the harmonics in Eq. (1): is $n=1$ or $n=-1$ the dominant one? The current results seem indicate that $n=1$ is the dominant one, which would be different than the convention usually used in our community. That requires to use an expression for Eq. (1) that is different from the authors’ one in the sense that it carries the sign of the gyrofrequency, the latter being evaluated as an absolute value (Albert, 2005).
- The authors should comment on the results of Cunningham et al. 2021, in particular their Figure 2 and related text.
- Figure 3 of Cunningham et al. 2021 indicates secondary peaks of high flux at higher L-shell than the NWC location which are similar to the one observed here at higher L-shells. This should also be cited.

References

Li, X., Selesnick, R., Mei, Y., O’Brien, D., Hogan, B., Xiang, Z., et al. (2024). First results from REPTile-2 measurements onboard CIRBE. *Geophysical Research Letters*, 51, e2023GL107521. <https://doi.org/10.1029/2023GL107521>

Liu, Y., Xiang, Z., Ni, B., Li, X., Zhang, K., Fu, S., et al. (2022). Quasi-trapped electron fluxes induced by NWC transmitter and CRAND: Observations and simulations. *Geophysical Research Letters*, 49, e2021GL097443. <https://doi.org/10.1029/2021GL097443>

Cunningham, G. S., Botek, E., Pierrard, V., Cully, C., & Ripoll, J.-F. (2020). Observation of high-energy electrons precipitated by NWC transmitter from PROBA-V low-Earth orbit satellite. *Geophysical Research Letters*, 47, e2020GL089077. <https://doi.org/10.1029/2020GL089077>

Peer Review History for 2024AV001354R

Reviewer #1

The authors have addressed my comments and I now find this paper acceptable for publication in AGU Advances.

Reviewer #2

[Reviewer Comments begin on the next page.]

Review for JGR Advances
Reference: 2024AV001354 (Xiang et al.)
Second Review : October 20 2024

The article “Earth-Based Transmitters Trigger Precipitation of Inner Radiation Belt Electrons: Unveiling Observations and Modeling Results” written by Xiang et al. and submitted to JGR Advances is about observations of electrons flux at LEO which present a typical “wisp” shape. Observations are made from CIRBE, a cube-sat built and managed by the authors’ team. This wisp has been observed in the past, first by Sauvaud et al. in 2008 from DEMETER. The novelty is the observation of multiple wisps. These wisps have been attributed to the NWC VLF transmitter signals emitted from Australia in the past. The authors explain the flux by modeling and simulation, with explanations that I had questioned in my last report and which the authors have answer in a very convincing manner.

First, the authors now write clearly they have used the FDC code developed at UCLA by Ni and Shprits in the 2005-2015 years for their computation. This code has been proven to be reliable and trusted since then. This is a relief. Some elements in the Supp. Mat. are minor enough they would let think the authors was learning and developing his model.

Second, the authors provide the 1D plots I had asked of the diffusion coefficients to better understand their claim, which otherwise were not understandable. In these plots newly added in the Supp. Mat. one can understand why they associate the diffusion to a particular harmonic. These figures can also be reproduced for verification purpose. These new results show conclusions can be different according to if the wave propagates in a ducted or unducted mode, as I had pointed out. The authors indeed concluded that too in their rebuttal, a text newly added in the Supp. Mat. The dissociation of the harmonics is also clearly shown while otherwise we were trusting the authors’ words without any demonstration.

Third, the authors had not explained enough in their original version the trapping duration (1 drift), the observed pitch angle (low pitch angle at the loss cone) and the initial condition vs pitch angle in a way that one could understand what type of diffusion is discussed and does produce the wisp signature. This is now clarified too.

Fourth, the authors had missed a few important references on this subject. Now, they are cited. As such the various improvements made during revision of this article are significant enough to me that the article becomes convincing and that it can be considered for publication in JGR advances.

Still, I see missing information or important corrections to make. I recommend for now major modification to be done before considering this article for publication. Details are given to the authors below.

Major information and references missing:

- 1- In the intro, relative to general modelling of NWC power and effects, please refer to Usanova et al. 2022.
- 2- Please mention the recent observations of wisp signals by ZH-1 in Zhao et al., 2019 (their figure 6, pink box).

3- All these studies listed below show diffusion coefficient of VLF transmitter waves with profiles that are peaked in pitch angle due to each of the resonance with VLF transmitter waves. This is known for a long while by any scientists who have done this type of simulations. However, this seems to be discovered here by the authors, reading either their description, with the absence of past references, or naïve explanations in their rebuttals. I ask these references are properly cited with a sentence acknowledging in the main text that fact, something like:

“Multiple past studies have shown that pitch angle diffusion coefficient for VLF transmitter waves have a typical shape composed of a sum of narrow peaks since each principal harmonic creates a peak of significant diffusion for a specific narrow pitch angle range depending on the values of (E, alpha, L) and of n the harmonic number (Abel and Thorne, 1998, 1999; Albert, 2012; Albert et al., 2016, 2020; Ripoll et al. 2014; Ross et al. 2019)”.

If the authors prefer to be more specific on where it can be seen in the above references, here is the list of figures: Figure 5 and 8 in Abel and Thorne, 1998; Figure 5 in Abel and Thorne, 1999; Figure 5-6 in Albert, 2012, Figure 1 in Albert et al., 2016; Figure 1-3 in Ripoll et al. (2014); Figure 2 and 5 in Ross et al. 2019; Figure 1 and 2 in Albert et al, 2020.

4- How do the wisp observations of the authors relate to the observation of zebra stripes in Ukhorskiy et al. 2015 and Lejosne and Roederer 2016. Please cite and discuss the similarities in the text.

5- In the introduction, about radiation belts, please mention the two recent reviews: Li and Hudson, 2019 and Ripoll et al., 2020

Corrections in the revised text and missing corrections

- 1- The other reviewer suggested to mention the work of Starks et al 2008 in which smaller amplitudes were computed for wave emitted from the ground. The authors followed his guidance. However, it turns out that Starks et al 2008 made a mistake in their computation using Helliwell attenuation profiles in trans-ionospheric propagation. This was found a few years later by Cohen et al., 2012. This point may have not be known by the other reviewer, and the authors. Therefore, I do not think that a mention to the article of Starks et al. 2008 should be made. If it was still chosen by the authors to refer to Stark et al., 2008, then a mention to the work of Cohen et al. 2012 should be made. I rather recommend not going into that direction and to just erase the following text: “**The smaller wave amplitude is estimated based on Starks et al. (2008), which may underestimate the transmitter intensity due to a broad illuminated area, and the peak intensity may be missed by spacecraft. In an updated study**”. Then, the authors can simply follow on with their sentence about the Stark et al. 2020 results.
- 2- Line 113 : “**Note that Ω doesn't include the sign of the charge so that N=1 is the principal resonance for whistler mode waves interacting with electrons.** “. Please indicate more clearly that this is not the convention used in Albert 2005 and Glauert and Horne 2005, which is used by most of the community. Please add the references. Please do not write contraction such as ‘doesn’t’.

- 3- Horne et al. 2005 is cited in the reference list but not in the main text. Please fix.

- 4- The authors wrote “Considering that the refraction index approaches infinity when the normal angles of highly oblique waves approach the resonance cone angle, we set the pitch angle diffusion coefficient to zero when the refraction index is larger than 200 (Ma et al., 2017). “. I agree with the authors on the method but I do not see that point being mentioned in Ma et al., 2017. Could the authors check? Also Mourenas, Artemev and co-authors have published a series of articles between 2013-2017 in which large refraction indices were considered to lead to large diffusion coefficients. In other words, highly oblique waves would lead to strong pitch angle diffusion. This was based on wrong approximative maths supported by a wrong code, as shown and discussed in a dedicated article (Albert, 2017). The correct result is: when refraction index approaches infinity (for highly oblique waves) the pitch angle diffusion coefficient tends to zero, as the authors do consider in their article. I recommend the authors cite the author who found the right result first. This is Albert, 2017. In simple words, I recommend to replace the citation Ma et al., 2017 by Albert, 2017.

- 5- Line 324: In “**In this study, we assume the same wave normal angles at different L values and only use different resonance orders ($N=-1,1,2$) to explain the observed three wisps.**”, it is unclear if ‘this’ refers to the authors’ study or Cunningham et al. Please use ‘our’ instead.

References

- Abel, B., & Thorne, R. M. (1998a). Electron scattering loss in Earth’s inner magnetosphere: 1. Dominant physical processes. *Journal of Geophysical Research*, 103(A2), 2385–2396. <https://doi.org/10.1029/97JA02919>
- Abel, B., & Thorne, R. M. (1998b). Electron scattering loss in Earth’s inner magnetosphere: 2. Sensitivity to model parameters. *Journal of Geophysical Research*, 103(A2), 2397–2407. <https://doi.org/10.1029/97JA02920>
- Abel, B., & Thorne, R. M. (1999). Correction to “Electron scattering loss in the Earth’s inner magnetosphere: 1, Dominant physical processes” and “Electron scattering loss in the Earth’s inner magnetosphere: 2, Sensitivity to model parameters” by Bob Abel and Richard M. Thorne. *Journal of Geophysical Research*, 104(A3), 4627–4628. <https://doi.org/10.1029/1998JA900121>
- Albert, J. M. (2005), Evaluation of quasi-linear diffusion coefficients for whistler mode waves in a plasma with arbitrary density ratio, *J. Geophys. Res.*, 110, A03218, doi:10.1029/2004JA010844.
- Albert, J. M. (2012), Dependence of quasi-linear diffusion coefficients on wave parameters, *J. Geophys. Res.*, 117, A09224, doi:10.1029/2012JA017718.

Albert, J. M., Starks, M. J., Horne, R. B., Meredith, N. P., & Glauert, S. A. (2016). Quasi-linear simulations of inner radiation belt electron pitch angle and energy distributions. *Geophysical Research Letters*, 43, 2381–2388. <https://doi.org/10.1002/2016GL067938>

Albert, J. M. (2017). Quasi-linear diffusion coefficients for highly oblique whistler mode waves, *J. Geophys. Res. Space Physics*, 122, 5339–5354, doi:10.1002/2017JA024124.

Albert, J. M., Starks, M. J., Selesnick, R. S., Ling, A. G., O'Malley, S., & Quinn, R. A. (2020). VLF transmitters and lightning-generated whistlers: 2. Diffusion of radiation belt electrons. *Journal of Geophysical Research: Space Physics*, 125, e2019JA027030. <https://doi.org/10.1029/2019JA027030>

Cohen, M. B., and U. S. Inan (2012). Terrestrial VLF transmitter injection into the magnetosphere, *J. Geophys. Res.*, 117, A08310, doi:10.1029/2012JA017992.

Glauert, S. A., and R. B. Horne (2005). Calculation of pitch angle and energy diffusion coefficients with the PADIE code, *J. Geophys. Res.*, 110, A04206, doi:10.1029/2004JA010851.

Lejosne, S., and J. G. Roederer (2016). The “zebra stripes”: An effect of F region zonal plasma drifts on the longitudinal distribution of radiation belt particles, *J. Geophys. Res. Space Physics*, 121, 507–518, doi:10.1002/2015JA021925.

Li, W., & Hudson, M. K. (2019). Earth's Van Allen radiation belts: From discovery to the Van Allen Probes era. *Journal of Geophysical Research: Space Physics*, 124, 8319–8351. <https://doi.org/10.1029/2018JA025940>

Ripoll, J.-F., J. M. Albert, and G. S. Cunningham (2014). Electron lifetimes from narrowband wave-particle interactions within the plasmasphere, *J. Geophys. Res. Space Physics*, 119, doi:10.1002/2014JA020217.

Ripoll, J.-F., Claudepierre, S. G., Ukhorskiy, A. Y., Colpitts, C., Li, X., Fennell, J., & Crabtree, C. (2020). Particle Dynamics in the Earth's Radiation Belts: Review of Current Research and Open Questions. *Journal of Geophysical Research: Space Physics*, 125, e2019JA026735. <https://doi.org/10.1029/2019JA026735>

Ross, J. P. J., Meredith, N. P., Glauert, S. A., Horne, R. B., & Clilverd, M. A. (2019). Effects of VLF transmitter waves on the inner belt and slot region. *Journal of Geophysical Research: Space Physics*, 124. <https://doi.org/10.1029/2019JA026716>

Usanova, M. E., Reid, R. A., Xu, W., Marshall, R. A., Starks, M. J., & Wilson, G. R. (2022). Using VLF transmitter signals at LEO for plasmasphere model validation. *Journal of Geophysical Research: Space Physics*, 127, e2022JA030345. <https://doi.org/10.1029/2022JA030345>

Ukhorskiy, A. Y., M. I. Sitnov, D. G. Mitchell, K. Takahashi, L. J. Lanzerotti, and B. H. Mauk (2014). Rotationally driven “zebra stripes” in Earth’s inner radiation belt, *Nature*, 507, 338–340.

Zhao, S., Zhou, C., Shen, X. H., & Zhima, Z. (2019). Investigation of VLF transmitter signals in the ionosphere by ZH-1 observations and full-wave simulation. *Journal of Geophysical Research: Space Physics*, 124, 4697–4709. <https://doi.org/10.1029/2019JA026593>

Peer Review History for 2024AV001354RR

Reviewer #2

[Reviewer Comments begin on the next page.]

Review for JGR Advances
Reference: 2024AV001354 (Xiang et al.)
Third Review : November 1 2024

The article “Earth-Based Transmitters Trigger Precipitation of Inner Radiation Belt Electrons: Unveiling Observations and Modeling Results” written by Xiang et al. and submitted to JGR Advances is about observations of electrons flux at LEO which present a typical “wisp” shape. Observations are made from CIRBE, a cube-sat built and managed by the authors’ team. This wisp has been observed in the past, first by Sauvaud et al. in 2008 from DEMETER. The novelty is the observation of multiple wisps. These wisps have been attributed to the NWC VLF transmitter signals emitted from Australia in the past. The authors explain the flux by modeling and simulation, with explanations that I had questioned in my first report and which the authors had answered in a very convincing manner in their first review. Though, there were still some inaccuracies left in the text. In their second review, they have now fixed the 5 major modifications which I had requested. There is still one minor modification that I asked below, about a corrected sentence which is still ambiguous and should be fixed. This can be simply done during the proof revision and I do not need to check it. Therefore, I am glad to recommend the publication of this article in JGR Advances.

Minor modification

1- Please replace ““Note that Ω does not include the sign of the charge so that $N=1$ is the principal resonance for whistler mode waves interacting with electrons. The convention form of the resonance condition can be found in previous studies (e.g. Albert, 2005; Glauert and Horne, 2005).”

By

“Note that Ω does not include the sign of the charge, as a convention in this study, so that $N=1$ is the principal resonance for whistler mode waves interacting with electrons, contrary to other studies for which $N=-1$ is the principal resonance (e.g. Albert, 2005; Glauert and Horne, 2005).”